# Anti-Obesity Effect of Liposomal Suspension and Extracts of *Hibiscus sabdariffa* and *Zingiber officinale* in a Murine Model Fed a Hypercaloric Diet

**DOI:** 10.3390/nu17142275

**Published:** 2025-07-09

**Authors:** Luis Edwardo Palomo-Martínez, Norma Paniagua-Castro, Gerardo Norberto Escalona-Cardoso, Diana E. Leyva-Daniel, Miguel A. A. Ibañez-Hernández, Yair Cruz-Narvaez, Liliana Alamilla-Beltrán

**Affiliations:** 1Instituto Politécnico Nacional, Departamento de Ingeniería Bioquímica, Escuela Nacional de Ciencias Biológicas, Av. Wilfrido Massieu 399, U. P. Adolfo López Mateos, Gustavo A. Madero 07738, Mexico; 2Instituto Politécnico Nacional, Departamento de Fisiología, Escuela Nacional de Ciencias Biológicas, Av. Wilfrido Massieu 399, U. P. Adolfo López Mateos, Gustavo A. Madero 07738, Mexico; 3Instituto Politécnico Nacional, Departamento de Bioquímica, Escuela Nacional de Ciencias Biológicas, Manuel Carpio y Plan de Ayala, Casco de Santo Tomás, Miguel Hidalgo 11340, Mexico; 4Instituto Politécnico Nacional, Departamento de Ingeniería Química, Escuela Superior de Ingeniería Químicas e Industrial Extractivas, Av. Luis Enrique Erro, 399, U.P. Adolfo López Mateos, Gustavo A. Madero 07738, Mexico

**Keywords:** obesity, polyphenols, *Hibiscus sabdariffa*, *Zingiber officinale*, liposomal suspension

## Abstract

Background/Objectives: Obesity is the primary risk factor for the development of chronic degenerative diseases. Multidisciplinary treatments target multiple pathologies associated with obesity. In this study, a potential adjuvant therapy was evaluated by combining extracts from Hibiscus *sabdariffa* and *Zingiber officinale*. These extracts were used in both a simple and liposomal suspension, the latter aimed at enhancing the activity of phenolic compounds and determining various metabolic benefits. Methods: In this research, the use of biotechnological approaches for the development of a liposomal suspension formulation with appropriate characteristics of stability, particle size, polydispersity index, concentration, and zeta potential induced an effective reduction in body weight and epididymal fat in a murine obesity model over 8 and 45 days. Results: Treatment with the liposomal suspension reduced variables in the lipid profile, aspartate aminotransferase activity, and energy expenditure, while also promoting an increase in locomotor activity. Conclusions: Therefore, it is suggested that the liposomal suspension represents an alternative for obesity treatment and the reduction of cardiovascular risks.

## 1. Introduction

*H. sabdariffa* (*Hibiscus sabdariffa roselle*) is a plant belonging to the *Malvaceae* family, with a high content of pharmacologically active compounds [1]. The main bioactive compounds responsible for therapeutic effects are organic acids and phenolic compounds such as anthocyanins (cyanidin-3-*O*-sambubioside and delphinidin-3-*O*-sambubioside); the latter has demonstrated a beneficial effect in the treatment of obesity [1,2]. Furthermore, *Z. officinale* (*Zingiber officinale roscoe*) is a perennial plant member of the *Zingiberaceae* family, which also contains phenolic bioactive compounds (gingerols, shogaols, and paradols) and terpenes (zingiberene, β-bisabolene, α-curcumene, α-farnesene, and β-sesquifelandrene). Among these, gingerols and shogaols are the most abundant and have demonstrated health and anti-obesity effects [3,4].

Polyphenols are secondary metabolites present in all vascular plants [5], that show low oral bioavailability due to extensive biotransformation mediated by enterocytes, liver, and gut microbiota [6]. Higher concentrations of polyphenols are required to obtain efficiency in vitro; however, if the administration is oral, only a small proportion of polyphenols can be absorbed under gastrointestinal conditions, due to acidic pH and enzymes, limiting their activity and possible health effects [5,7].

There are numerous studies regarding the use of technologies that allow the biological activity of phenolic compounds to be preserved, including the encapsulation of Hibiscus sabdariffa extracts, with beneficial effects in the treatment of obesity and other pathologies. The extract encapsulated by spray-drying reduced glucose levels and total cholesterol [8]; the consumption of encapsulated hibiscus extracts reduced body weight, BMI, body fat, waist-to-hip ratio, and improved liver steatosis disease [9]. Although there is no evidence of ginger encapsulation processes, the anti-obesity effect of ginger extract has been studied. The sustained activation of the peroxisome proliferator-activated receptor δ (PPARδ) pathway with ginger extract attenuated diet-induced obesity and improved exercise endurance capacity by increasing skeletal muscle fat catabolism. In diets supplemented with ginger, a decrease in body weight, hepatic steatosis, and low-grade inflammation, as well as an improvement in insulin resistance, were observed [10]. The ginger extract in water showed an effect on reducing body weight gain at the transcriptional level of proteins that metabolize energy [11].

The specific use of biotechnologies such as oral liposomes with polyphenols allows their encapsulation and protection against inactivation and degradation until their consumption or administration [5,12]. Compared to traditional drug delivery systems, liposomes exhibit improved properties, stability, biodistribution in target organs, and reduced barriers to cellular and tissue absorption and systemic toxicity [13,14,15]. Liposomes are spherical vesicles composed of one or more phospholipid bilayers. Their amphiphilic chemical composition allows them to encapsulate hydrophilic biomolecules and/or drugs in the aqueous core and lipophilic biomolecules and/or drugs in the lipid bilayer [16,17]. Depending on the composition of the formulations, properties such as size, zeta potential, and stiffness of the liposomes can be controlled [18]. There are several methods to obtain liposomes, such as thin-layer hydration, ethanol injection, reverse-phase evaporation, and others such as membrane extrusion, sonication, and freeze-thawing [19]. Therefore, in this study, liposomes with extracts from *H. sabdariffa* and *Z. officinale* were synthesized, characterized, and analyzed, and their effects were compared with non-encapsulated extracts using a murine model of obesity induced by a hypercaloric diet.

## 2. Material and Methods

### 2.1. Extracts Preparation

The calyxes of *H. sabdariffa* (“Ñanirey El Guerrero”, Mexico), acquired in a local market were frozen at −18 °C for 18 h and freeze-dried (FreeZone 18 Lyophilizer, Labcono, Kansas City, MO, USA) for 8 h at −49 °C and 0.12 mBar to ensure complete drying of the product. It was then stored at −18 °C in plastic containers for further analysis [20]. An extract of *H. sabdariffa* was prepared by adding 5 g of lyophilized powder to 100 mL of hot distilled water (80 °C) for 10 min. After that, constant stirring was applied at 600 rpm for 3 h, followed by centrifugation at 4170× *g* for 10 min at 4 °C and then filtered with Whatman no. 2 filter paper.

For the *Z. officinale* extract, 10 g of commercial dried organic powder (Pragná, Orgániks, México) acquired in a local market, were added to 100 mL of an ethanol/water solution (25:75), under constant stirring at 600 rpm for 3 h. It was then centrifuged at 4170× *g* for 10 min at 4 °C and filtered with Whatman no. 2 filter paper. Finally, the ethanol was evaporated using a rotary evaporator (B-490, Büchi, Flawil, Switzerland), and the residue was resuspended in water to the initial volume.

For the treatment of the mice, the extracts of *H. sabdariffa* and *Z. officinale* were mixed in a 1:1 ratio for the acute treatment and in a 2:1 ratio for the subchronic treatment. In total, 35 mL was placed in the mice’s drinker and the consumption of the treatment was measured daily.

### 2.2. Determination of Total Phenolic Compound Content

The concentration of total phenolic compounds (TPC) was determined using the Folin Ciocalteu method with slight modifications as described by [21].

### 2.3. Untargeted Phytochemical Profiling Using FTICR-MS

Metabolite detection was performed using a Bruker 7T SolariX XR Fourier transform ion cyclotron resonance mass spectrometer (FTICR-MS, Bruker Daltonik, GmbH, Bremen, Germany). Samples were introduced into the electrospray source by direct infusion using a 50 μL syringe (Hamilton Company, Reno, NV, USA) at a flow rate of 2 μL/min. The electrospray source was operated with the following settings: a capillary voltage of 4.5 kV, a drying gas temperature of 180 °C, and a flow rate of 4 L/min. The analysis was performed in positive and negative ionization mode with a mass range of 43–2000 Da. For bulk analysis, 100 scans were accumulated, resulting in a total acquisition time of 2 min and a resolving power of 2,000,000 at *m*/*z*.

### 2.4. Preparation of Liposomal Suspension

Liposomes were obtained based on the methodology employed by Dag & Oztop [22] with some modifications. Soy lecithin (Margarita Naturalmente, Mexico City, México) at 1% (*w*/*v*) was added to homogenize the lipids on the flask walls. Then, Tween 80 (Sigma-Aldrich, Merck KGaA, Darmstadt, Germany) at 0.16% (*w*/*v*) was added as a surfactant, followed by the combination of extracts. Immediately, the mixture was homogenized at 20,000 rpm for 2 min using an Ultra-Turrax device (T18 basic, IKA Works, Wilmington, NC, USA). Subsequently, it underwent probe-type sonication 130-Watt Ultrasonic Processors, (Cole-Parmer, Vernon Hills, IL, USA) using a TT13 probe at 75% amplitude for 12 min (6 cycles of 30 s on and 30 s off).

### 2.5. Average Particle Size and Zeta Potential

The liposomal suspension sample was diluted in distilled water (1:100 μL), and homogenized, and then the average particle size (nm) and polydispersity index (PDI) were determined using dynamic light scattering. The zeta potential (mV) of the samples was measured using laser Doppler microelectrophoresis (Zetasizer Nano ZEN3600, Malvern Instrument, Worcestershire, UK) [23].

### 2.6. Encapsulation Efficiency of Phenolic Compounds

To determine the encapsulation efficiency percentage (% *EE*), the methodology employed by [23] was utilized with some modifications. The obtained liposomes were centrifuged at 21,370× *g* for 30 min. The precipitate was collected, and the liposomal bilayers were disrupted using 0.02% Triton X-100 (Sigma-Aldrich, Merck KGaA, Darmstadt, Germany) in distilled water. One milliliter was added, and the mixture was vortexed for 4 min to homogenize and release the encapsulated extract. The % *EE* was calculated with the following equation:(1)$$\%\;EE=\frac{P}{\left(S+P\right)}\times 100$$
where *P* is the content of encapsulated phenolic compounds in the liposomes (after disruption by Triton X-100), and *S* is the amount of free phenolic compounds in the supernatant.

### 2.7. Particle Concentration in the Liposomal Suspension

Particle concentration was determined using a Nanoparticle Tracking Analysis (NTA) instrument (Nanosight, NS300, Malvern Panalytical, Worcestershire, UK) equipped with the software NTA 3.2 Dev Build V. 3.2.16 (Malvern WR14 1XZ, Malvern Panalytical, Worcestershire, UK). A total of 1 mL of each diluted (1:1000 in distilled water) sample was analyzed. The number of liposomes was obtained using the following configuration: camera type: sCMOS, level 9 (NTA 3.0 levels), blue488 laser, slider, and gain: 607, shutter: 15 ms, frame rate: 25 FPS, syringe pump speed: 25 arbitrary units, temperature: 21.0–21.2 °C, and viscosity: (Water, Millipore, Burlington, MA, USA) 0.971–0.975 cP [24].

### 2.8. Confocal Laser Scanning Microscopy (CLSM)

Confocal laser scanning microscopy (CLSM) of the liposomal suspensions was performed on an LSM 710 NLO microscope (Carl Zeiss, Oberkochen, Germany). The samples were processed as follows: 25 μL of the liposomal suspension + 75 μL of deionized water + 15 μL of 0.1% Nile Red (Sigma Chemical Co., St. Louis, MO, USA), with a rest period of 10 min. Subsequently, 5 μL of the sample was placed on glass slides and observed with two laser lines (488 nm and 633 nm with a capacity of 6.5%). Images were taken with a 63×/1.40 Plan-Apochromat Oil Dec M27 objective with a resolution of 512 × 512 pixels^2^ (0.19 μm^2^ pixel size).

### 2.9. Evaluation of the Murine Model

Forty male CD1 mice (PROPECUA S.A. DE C.V., Mexico City, Mexico) with an initial weight of 25 to 30 g were used. Mice were individually housed in cages with sawdust bedding under controlled temperature conditions (22 ± 1 °C), with 12 h of light/dark cycles, with food and water ad libitum. The experiment was approved by the Bioethics Committee (Escuela Nacional de Ciencias Biologicas/IPN, code CEI/029/2019, 20 December 2019), and all experiments complied with the guidelines in accordance with the UK Animals (Scientific Procedures, London, UK) Act 1986 and associated guidelines, EU Directive 2010/63/EU for animal experiments.

A group of 10 mice was fed with a standard diet (STD) (control group), and the other 30 with a hypercaloric diet (HCD) (Table 1 and Table 2) for three weeks. After that time, the HCD-fed mice group was divided into three; the terminology used starts with the treatment, as shown in Table 3. The intervention with the treatment was divided into three phases, the initial acute phase, and the chronic follow-up phase. The initial phase consisted of (1) 8 days with the high concentration of *Z. officinale* (total concentration 75 mg/mL) and (2) three weeks of rest with the same assigned diet and *ad libitum* access to water. Subsequently, (3) the chronic follow-up treatment phase was reinstated for 45 days with half the concentration of *Z. officinale* extract, maintaining the concentration of the *H. sabdariffa* extract (total concentration of 50 mg/mL) (Figure 1, Table 3). The body weight (g), water intake and/or treatment (mL/day), food intake (g) and consumed calories were taken every four days in the initial phase and every five days in the follow-up phase. The treatment was consumed on demand to avoid stressing the mice, and it was prepared and changed every 2 days. The body weight gains or losses were determined using the final weigh–initial weigh (g), and the relative fat pad weight was determined using the formula:(2)$$\mathrm{RFPW}\;(\%)=\frac{Average\;fat\;pad\;weigh\;of\;group\;(g)}{average\;body\;weight\;of\;group\;(g)}\times 100$$

The open field test followed the protocol by [25] at the end of the treatment, in which the moved squares (5 × 5 cm) were measured for 5 min in a rectangular enclosure that prevented escape, thus determining the general activity of the mice. The recorded data were then compared.

Furthermore, at the conclusion of the treatment period, all groups were fasted for 12 h to obtain a blood sample, it was taken and centrifuged at 4000× *g* for 10 min at 5 °C to obtain the serum. The concentrations of total cholesterol (TC), high-density lipoprotein cholesterol (HDL-C), low-density lipoprotein/very-low-density lipoprotein cholesterol (LDL-C/VLDL-C), triglycerides (TG), aspartate aminotransferase (AST), and alanine aminotransferase (ALT) were determined using UV–Vis spectrophotometric assays (Multiskan 60, Thermo Scientific, Vantaa, Finland) using commercial kits (Sigma-Aldrich, Darmstadt, Germany and Randox, Crumlin, County Antrim, UK). Additionally, the atherogenic index was determined [26] using the formula:(3)$$Atherogenic\;Index=\frac{TC}{\mathrm{HDL-C}}$$
where HDL-C is high-density lipoprotein cholesterol, and TC is total cholesterol in plasma.

They were then sacrificed by cervical dislocation, and epididymal adipose tissue samples were collected through a lower abdominal incision and then weighed [27].

### 2.10. Statistical Analysis

The results obtained were analyzed with the statistical program (GraphPad Prism version 6.0 Software, Inc., San Diego, CA, USA). To compare the effects of the extract and liposomal suspension in murine models, one-way ANOVA was performed, followed by a Tukey’s multiple comparisons post hoc test (* = *p* ≤ 0.05, ** = *p* ≤ 0.01, *** = *p* ≤ 0.001). The data of biochemical analysis represent the mean ± standard error; different letters indicate significant differences between groups, Tukey’s post hoc (*p* ≤ 0.05).

## 3. Results and Discussion

### 3.1. Characterization of Treatments

An FTICR-MS analysis of the *H. sabdariffa* extract identified a total of 23 phenolic compounds, among which four of the most important with pharmacological effects included chlorogenic acid, cyanidin-3-sambubioside, and delphinidin-3-*O*-glucoside (Table 4) [1]. The constituents are highly variable according to sensitive analytical methods; however, in other studies using a high-performance pressure chromatography (HPLC) system and LC-ESI-MS, the same molecules were found [28,29]. Meanwhile, an analysis of the *Z. officinale* extract identified a total of 24 of the phenolic compounds, the highlighted compounds being 6-gingerol, 6-shogaol, 6-paradol, and 6-gingerdiol (Table 5) [3]. Our results are consistent with other studies in which the molecules were found by LC-MS/MS [30]. In our results, the presence of the most important phenolic compounds was corroborated for both extracts. However, it is important to consider that new compounds are described that should be taken into account in future research.

Table 6 shows the results obtained from the characterization of the liposomal suspension, including the total phenolic compounds in the extract, particle size, PDI, Z-potential, % encapsulation, and concentration. Meanwhile, the liposome formation was confirmed by confocal microscopy (Figure 2).

The conditions used for each of the extracts were adequate and share similarities with other studies. In a study comparing *H. sabdariffa* extractions with cold (25 °C) and hot (90 °C) water, it was found that total phenolics were better extracted with hot water (90 °C/16 min) [31,32], and achieved a better therapeutic effect in metabolic diseases [32,33]. Furthermore, for the *Z. officinale* extract the use of ethanol/water shared similarities with the work described by Jan et al. [34] by showing a higher concentration of total phenolic compounds.

Particle size, polydispersity index (PDI) and zeta potential are characteristic parameters for liposomal formulations [35]. Some researchers have reported that the bioavailability of molecules encapsulated in liposomes directly depends on their particle size; for example, particles with smaller sizes (≤400 nm) have approximately three times more bioavailability compared to larger ones [36]. In this work, suitable liposome sizes with an optimal polydispersity index were obtained, indicating high liposome homogeneity, desirable for oral administration systems. Achieving high homogeneity with polydispersity index (PDI) values < 0.3 is a challenge to the physical characteristics of a liposomal system [37]. Furthermore, the stability of liposomal suspensions depends largely on the surface charge or zeta potential, which can be estimated from their electrophoretic mobility and potential correlates with the physical stability of a liposomal system, where a high zeta potential (greater than +35 mV or less than −35 mV) leads to a more stable suspension [38,39]. The liposomal suspension obtained in this research did not show these values; however, it maintained its characteristics up to 28 days, which could be due to the use of Tween 80 as a surfactant, which contributed to the stability of the suspensions [40]. At 0, 7, 21, and 28 days, particle size was measured (153.46 ± 0.35, 160.93 ± 6.76, 190.26 ± 3.62, and 217.86 ± 1.06, respectively) maintaining the IPD (<0.4) and Z potential (initial from −26.2 ± 0.62 to −25.13 ± 0.15 at 28 days); these data show the stability of the suspension due to the small changes observed. Regarding the concentration of liposomes obtained in the suspension, it was high due to the interaction of both hydrophilic and lipophilic compounds from both extracts, which may be incorporated into the interior and bilayer of the liposomes [41]. Similar results were found by Gibis et al. [38], where liposomes were obtained from lecithin (1% *w*/*w*) using the microfluidization method with grape extract rich in polyphenols. An initial particle size of 86.5 nm, PDI below 0.35, and an encapsulation efficiency of 99.5 ± 4.7% were observed. In another study by Guldiken et al. [42], liposomes loaded with carrot extract were obtained with a particle diameter of 44.7 nm. The variations in the characteristics of the liposomes obtained in this research compared to other studies depend entirely on the type of phospholipids, the phenolic compounds present, and the methods used for liposome preparation. Nevertheless, all presented desirable ranges for clinical use.

### 3.2. Obesity Murine Model

For the acute period of 8 days, the results of the fluid intake of the treated mice are shown in Figure 3, showing a significant difference between all groups except between HCD+E and HCD+L, with the control group showing a higher water intake with 13.2 ± 0.9 mL. Regarding caloric intake, at the end of 8 days (Figure 4), a significant difference was only observed between the HCD + E and HCD + L groups (19.1 ± 0.7 and 22.3 ± 0.4 cal, respectively). In Figure 5, the HCD + V group showed a significant weight gain compared to the STD group, while the groups receiving the extract and liposomal suspension showed a significant loss of body weight (>20%) compared to the other groups (Table 7). The HCD + L and HCD + E treatments did not differ from each other. The evaluation of the efficacy of a new anti-obesity treatment is to demonstrate a 5–10% body weight loss effect over six months [43]. A sustained weight loss of more than 10% overall body weight improves many of the complications associated with obesity (e.g., the prevention and control of type 2 diabetes, hypertension, fatty liver disease, and obstructive sleep apnea), as well as quality of life [44], so the groups of mice receiving the treatments reduced more than 20% of their body weight over a very short period (8 days), an effect that exceeded the established standards.

However, it was considered that the liposomal treatment showed a better effect when compared to the extract, since the intake treatment (mL), foods consumed (g), and the overall average calories consumed were minor, and consequently, a decrease in the negative effects of ingesting a high-calorie diet was observed.

Therefore, the administration of treatments was paused for two weeks to allow the mice to stabilize and recover (maintaining the same diet and ad libitum water). After this period, the concentration of the *Z. officinale* extract was reduced, and the concentration of *H. sabdariffa* was maintained, to be administered for 45 days.

Over this period, there was no difference in liquid intake between the STD control, HCD + V, and HCD + L groups. However, the HCD + L group decreased fluid consumption compared to the other 3 groups (Figure 6).

Regarding caloric consumption (Figure 7), the HCD + L group consumed significantly more calories than the other groups. Additionally, regarding the average weight gain or loss at the end of the 45-day period, an increase was observed for the STD control and HCD + V groups (2.3% and 2%, respectively). Even though there was no significant difference between the two groups, the diet caused different metabolic effects, with HCD + E and HCD + L showing a decrease of 13.2% and 3%, respectively (Table 7).

Based on these results, the weight loss during the treatment was attributed to reduced calorie intake (lower food consumption), observed only in the HCD + E group. While for the HCD + L group, the opposite was observed. This group had the highest caloric intake, significantly different from the other groups, and the result could suggest that the weight loss effect was due to the greater energy expenditure generated by the treatment intake.

To determine the physical activity of the animals, the locomotor activity test was performed in the open field (Figure 8). Here, the HCD + L group showed significantly higher activity compared to the other treatments, which could indicate that the mice had a higher energy expenditure [45]. A diet rich in fats and sugars reduces locomotor activity in mice in as little as 3–5 h before significant weight gain occurs, this decrease is due to rapid changes in motivational and reward brain circuits, which affect exploratory behavior [46]. Thus, the early reduction in activity contributes to the accelerated development of obesity. In addition, it has been shown that the inhalation by mice of ginger, thyme, peppermint and cypress oils produced a reduction in immobility of between 5% and 22%, which could explain the stimulant and antidepressant effects observed [47]. Our findings align with those of Totten et al. [48], who observed a reduction in motor activity in mice fed a high-fat diet (60% cal) compared to those fed a low-fat diet (10% kcal) over a 16-week period. Concerning body weight (Figure 9), the HCD + extract and HCD + L groups significantly decreased their body weight compared to the HCD + V group. The epididymal fat is a metabolically active abdominal fat depot widely used to study adipose tissue biology in rodents [49].

Relating to the results of epididymal fat and RFPW % (Figure 10 and Table 8), a significant increase was observed in the HCD + V group, while in the HCD + E and HCD + L groups, the increase was lower, although not different from HCD + V. The hypercaloric diet promoted an increase in visceral fat deposits, which were observed in a lower amount in the STD control group, followed by HCD + E, HCD + L, and HCD + V (0.49 ± 0.08, 0.73 ± 0.17, 1.02 ± 0.11, and 1.19 ± 0.15 g, respectively). There was a significant difference between the control group and HCD + V and HCD + L. However, the trend towards fat reduction in the HCD + L group was visualized.

The results of the biochemical profile of the animals from different treatments are shown in Table 8.

It is evident that the hypercaloric diet used in this study contains a significant amount of fructose, known to elevate metabolic markers such as a blood glucose [26]. This effect was consistently observed across all experimental groups with HCD, with no significant modulation attributable to the administered treatment. Furthermore, no effect on trygliceride concentration was observed, with no significant differences between the experimental groups. Total cholesterol (TC) was significantly higher in the HCD + L group and was like HCD + V, while the HCD + E group had a smaller increase compared to those groups. Elevated serum concentrations of total cholesterol (TC) are associated with a high risk of cardiovascular diseases (CVD) [50]. The normal concentration of total cholesterol in mice is ≤130 mg/dL and in humans ≤ 200 mg/dL. In this research, the HCD diet increased total cholesterol in all three groups of mice. Although, for the HCD + E group, there was a trend of lower concentrations compared to HCD + V (Table 8).

Moreover, the relationship between elevated concentrations of low-density lipoprotein cholesterol (LDL) and the risk of atherosclerotic cardiovascular disease is very close [51]. Regarding LDL/VLDL concentrations, a lower concentration was observed in the HCD + L group, followed by HCD + E and HCD + V. The liposomal suspension decreased LDL/VLDL levels compared to the HCD + V group (Table 8).

High-density lipoprotein cholesterol (HDL) is inversely associated with the risk of vascular complications and is considered an anti-atherosclerotic lipoprotein [50]. In this study, a higher concentration of HDL was observed in the HCD + L group compared to STD and HCD + V, with the latter being even lower (Table 8). This result is of utmost importance, as despite the HCD diet’s ability to decrease HDL and increase LDL/VLDL, the liposomal suspension treatment counteracted this effect, representing a protective effect on the endothelium with anti-inflammatory capacity [52]. Additionally, the atherogenic index was closer to the STD control mice group. The atherogenic index remarkably increased in the HCD + V group, while it decreased in the HCD + E and HCD + L groups compared to the HCD + V group (Table 8).

Regarding aspartate aminotransferase (AST) and alanine aminotransferase (ALT), they are the most sensitive indicators for diagnosing liver cell damage, so an increase in serum concentrations of AST and ALT can reflect a degree of liver cell damage [53]. Normal concentrations of AST in mouse serum are within the range of 50–100 U/L [54], and all studied groups were within this range. However, the liposomal suspension significantly decreased AST concentrations, suggesting an additional hepatoprotective effect. All the same, there was no significant difference in serum ALT concentrations between the studied groups.

*H. sabdariffa* and *Z. officinale* show a favorable safety profile in both animal and human studies. In rodents, hibiscus shows a high LD_50_ (>2000–5000 mg/kg), and in humans no significant adverse effects have been reported up to doses of approximately 10 g/day or extracts of 300 mg/kg/day [55]. In addition, hibiscus has demonstrated beneficial effects on lowering blood pressure and blood lipid levels. Meanwhile, studies in rats indicate that oral doses of ginger up to 2000 mg/kg/day are generally safe, while in humans habitual consumption of 2–4 g/day is well tolerated, with mild, mainly gastrointestinal side effects [56]. Ginger has shown anti-inflammatory properties, antioxidant properties, and positive effects in relieving nausea and digestive problems. These data suggest that the clinical uses of both products can be extrapolated to human use without significant risks of toxicity when therapeutic doses are respected.

Drugs approved for the treatment of obesity act through specific mechanisms. These include the inhibition of fat absorption (orlistat), the activation of GLP-1 receptors to suppress appetite (liraglutide and semaglutide), and stimulation of the central nervous system to reduce hunger (phentermine). However, these drugs can cause relevant adverse effects, such as gastrointestinal disorders, and cardiovascular and neuropsychiatric effects. This limits their short- or medium-term use under strict medical supervision. A recent analysis of 132 randomized clinical trials involving 48,209 participants found that the combination of phentermine and topiramate was the most effective in weight reduction, followed by GLP-1 agonists, with semaglutide showing the greatest benefit in terms of weight loss and the risk of adverse effects [57].

In the present study, an effective reduction in body weight was observed with the administration of the simple extract combination (13.2%) and with the liposomal suspension (3%) over a period of 45 days with a hypercaloric diet. In addition, in the context of the metabolic effect, the liposomal suspension treatment had better results compared to the other groups in terms of LDL/VLDL reduction and HDL increase, and this group of mice also remained more active and consumed more calories, suggesting that, regardless of the damage caused by the DHC diet intake, the liposomal suspension had the capacity to attenuate its effects. As previously mentioned, coupled with the benefits of phenolic compounds present in hibiscus and ginger extracts, the use of oral liposomes in suspension form as an encapsulation medium is a specific alternative to enhance the effectiveness of various bioactives, which facilitate the arrival of bioactives to target organs and improve treatment performance [19].

Currently, there is no evidence of the use of the combination of *H. sabdariffa* and *Z. officinale* extracts with anti-obesity effect, and even less with the use of liposomal formulations. There is only one study by Ahad et al. [58] in which both extracts were administered with losartan to hypertensive rats. Treatment significantly reduced systolic and diastolic blood pressure. However, there is great variability in studies of the extracts individually, with different models, doses, treatment times, and results obtained, of which stand out the use of high doses and with therapeutic effects as in decreased food intake, decreased lipogenesis, increased lipolysis, stimulation of β-oxidation of fatty acids, inhibition of adipocyte differentiation and growth, attenuation of inflammatory responses and suppression of oxidative stress [11,59,60,61]. The difference with the present study was that both extracts were mixed and formed liposomes for administration. The final effect was a decrease in body weight and some variables of the metabolic syndrome, in addition to reducing the liver enzyme AST, making it a potential hepatoprotective agent.

## 4. Conclusions

The obtained liposomal suspension showed optimal initial characteristics of stability, particle size, polydispersity index, phenolic compound, encapsulation percentage, particle concentration per milliliter, and zeta potential.

Both treatments, suspension and liposomes, reduced body weight in the murine model of obesity during 8 and 45 days of treatment. However, the liposomal suspension treatment showed a better reduction of body weight, LDL/VLDL and AST, and an increase in HDL-C, thus being considered a potential alternative or complementary treatment for obesity.

## Figures and Tables

**Figure 1 nutrients-17-02275-f001:**
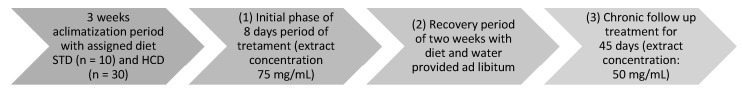
Experimental design.

**Figure 2 nutrients-17-02275-f002:**
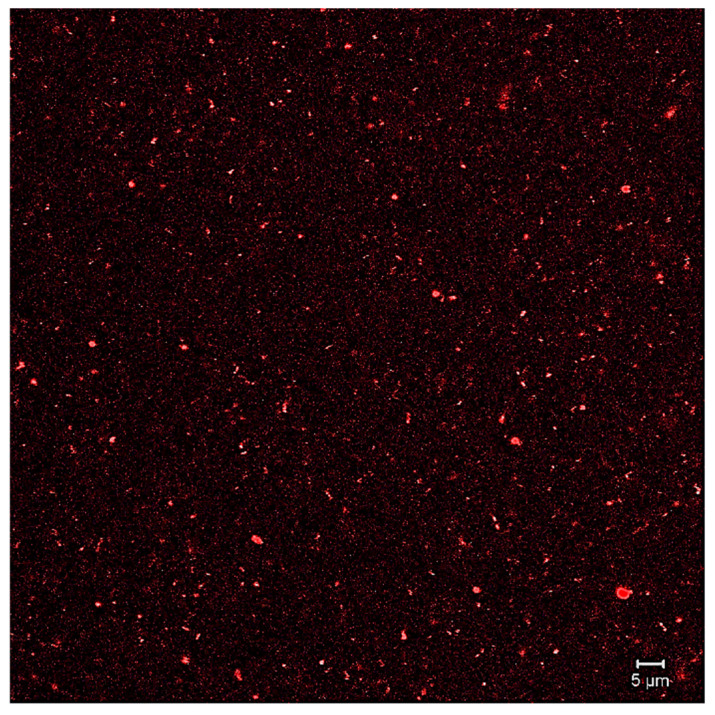
Image of CLMS.

**Figure 3 nutrients-17-02275-f003:**
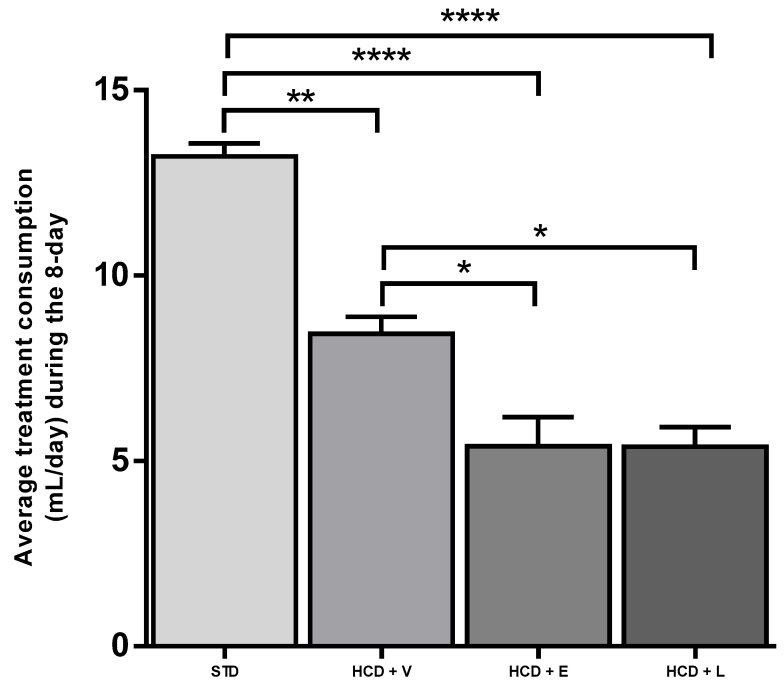
Average treatment consumption during the 8-day acute treatment. STD: standard diet, HCD + V: hypercaloric diet plus vehicle, HCD + E: hypercaloric diet plus extract, HCD + L: hypercaloric diet plus liposomal suspension. ANOVA Tukey post hoc * *p* < 0.05, ** *p* < 0.01, **** *p* < 0.0001.

**Figure 4 nutrients-17-02275-f004:**
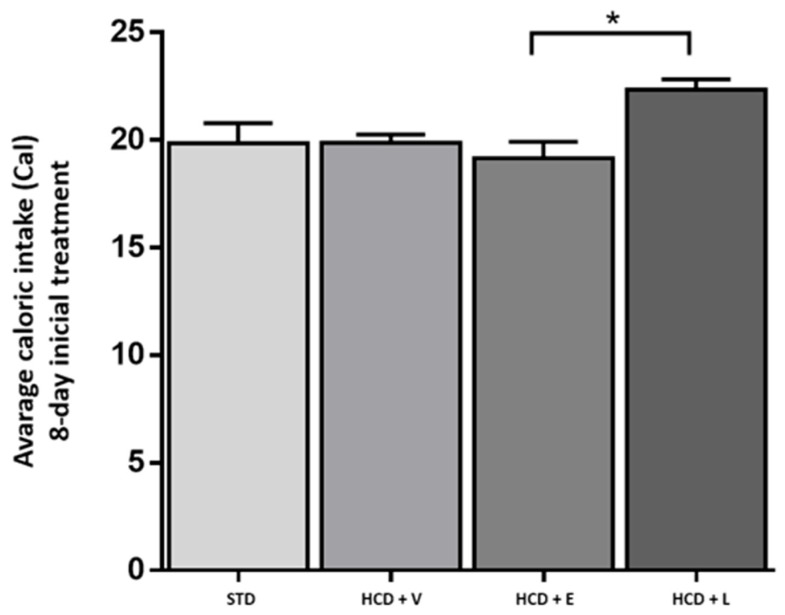
Average caloric intake during the 8-day acute treatment. STD: standard diet + water, HCD + V: hypercaloric diet plus vehicle, HCD + E: hypercaloric diet plus extract, HCD + L: hypercaloric diet plus liposomal suspension. ANOVA one-way, Tukey post hoc * *p* < 0.05.

**Figure 5 nutrients-17-02275-f005:**
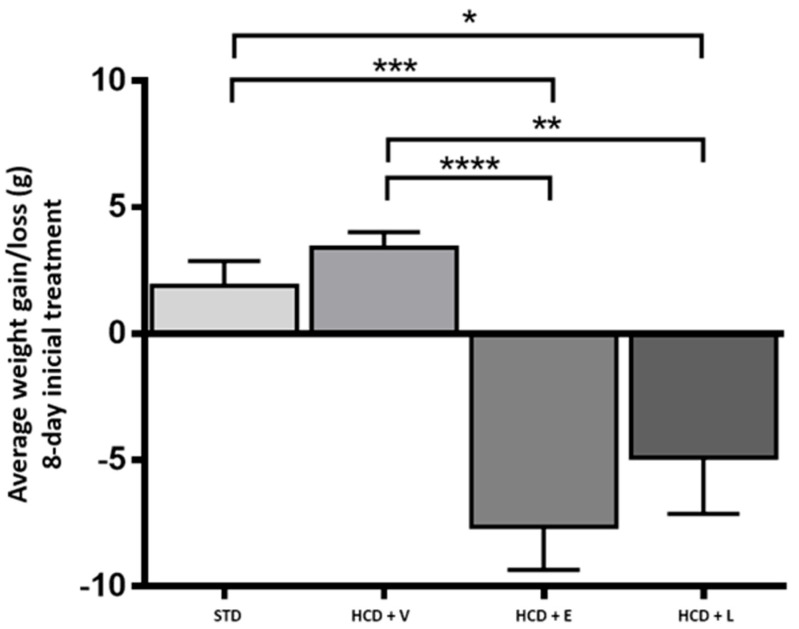
Average weight gain/loss during the 8-day acute treatment. STD: standard diet, HCD + V: hypercaloric diet plus vehicle, HCD + E: hypercaloric diet plus extract, HCD + L: hypercaloric diet plus liposomal suspension. One-way ANOVA, Tukey post hoc * *p* < 0.05, ** *p* < 0.01, and *** *p* < 0.001, **** *p* < 0.0001.

**Figure 6 nutrients-17-02275-f006:**
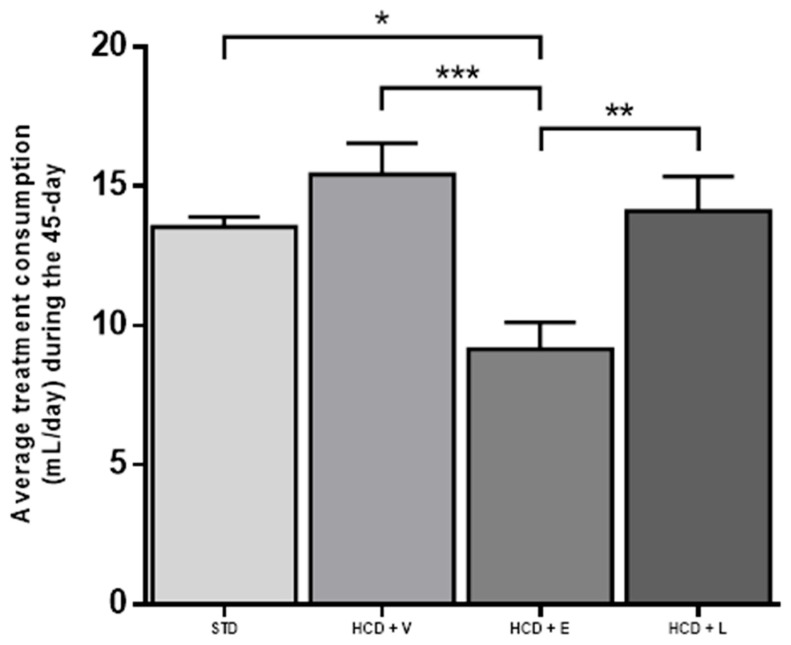
Average treatment consumption during the 45-day period. STD: standard diet + water, HCD + V: hypercaloric diet plus vehicle, HCD + E: hypercaloric diet plus extract, HCD + L: hypercaloric diet plus liposomal suspension. One-way ANOVA, Tukey post hoc, * *p* < 0.05, ** *p* < 0.01, and *** *p* < 0.001.

**Figure 7 nutrients-17-02275-f007:**
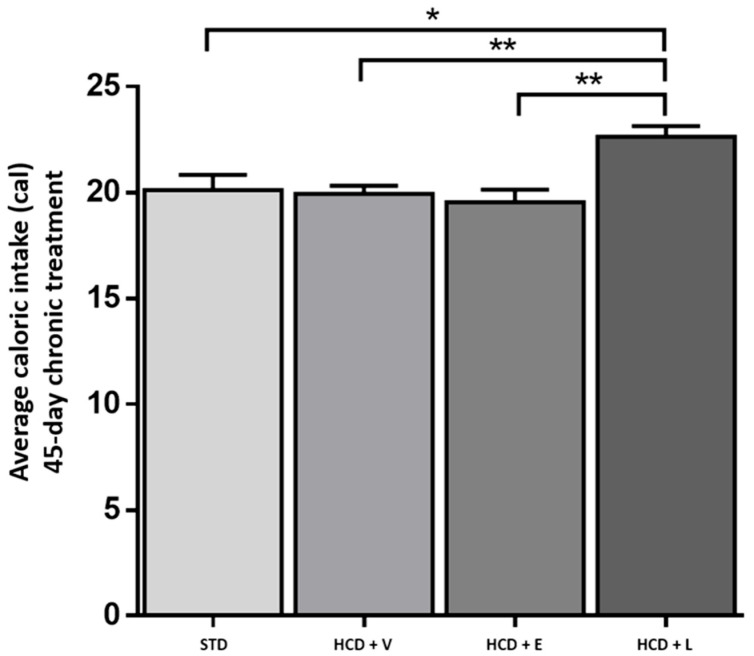
Average caloric intake during the 45-day chronic treatment. STD: standard diet, HCD + V: hypercaloric diet plus vehicle, HCD + E: hypercaloric diet plus extract, HCD + L: hypercaloric diet plus liposomal suspension. One-way ANOVA, Tukey post hoc, * *p* < 0.05, ** *p* < 0.01.

**Figure 8 nutrients-17-02275-f008:**
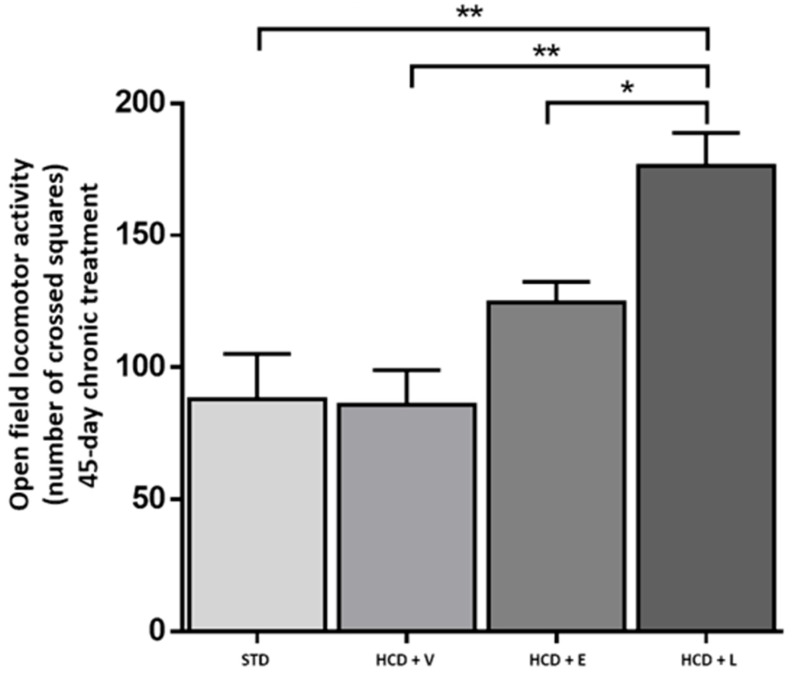
Open field locomotor activity at the end of the 45-day chronic treatment. STD: standard diet, HCD + V: hypercaloric diet plus vehicle, HCD + E: hypercaloric diet plus extract, HCD + L: hypercaloric diet plus liposomal suspension. One-way ANOVA, Tukey post hoc, * *p* < 0.05, ** *p* < 0.01.

**Figure 9 nutrients-17-02275-f009:**
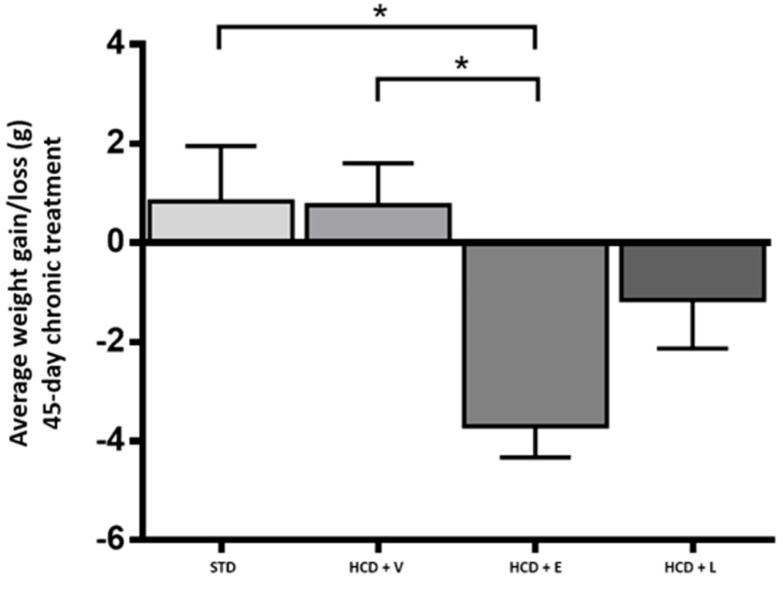
Change in body weight at the end of the 45-day chronic treatment. STD: standard diet, HCD + V: hypercaloric diet plus vehicle, HCD + E: hypercaloric diet plus extract, HCD + L: hypercaloric diet plus liposomal suspension. One-way ANOVA, Tukey post hoc, * *p* < 0.05.

**Figure 10 nutrients-17-02275-f010:**
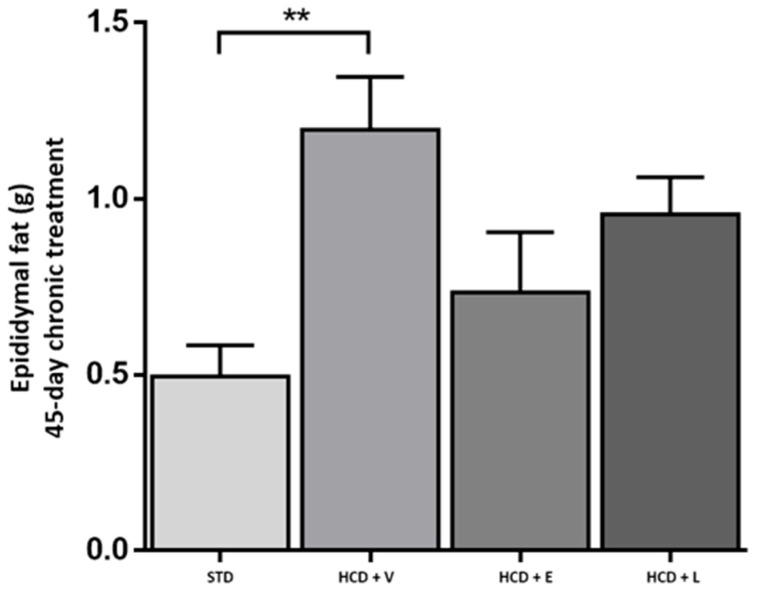
Epididymal fat in animals at the end of the 45-day chronic treatment. STD: standard diet, HCD + V: hypercaloric diet plus vehicle, HCD + E: hypercaloric diet plus extract, HCD + L: hypercaloric diet plus liposomal suspension. One-way ANOVA, Tukey post hoc, ** *p* < 0.01.

**Table 1 nutrients-17-02275-t001:** Nutritional composition of the diets used for mice.

Macronutrients	Standard Diet STD	Hypercaloric Diet HCD
Proteins	0.909 cal/g (28.6%)	0.841 cal/g (17.5%)
Carbohydrates	1.844 cal/g (58%)	0.409 cal/g (40.9%)
Lipids	42.61 cal/g (13.4%)	2 cal/g (41.6%)
Total energy content	3.18 cal/g	4.81 cal/g

**Table 2 nutrients-17-02275-t002:** Hypercaloric diet (HCD) composition.

Ingredients	Concentration (g/kg)
Sucrose	340
Butter	210
Casein	195
Corn starch	140.5
Cellulose	50
Mineral mix (AIN-93 G-MX)	43
Vitamin mix (AIN-9-VX)	15
DL—methionine	3
Choline	2
Cholesterol	1.5

**Table 3 nutrients-17-02275-t003:** Studied groups of mice.

Group	Diet	Treatment
STD	Standard	Water
HCD + V	HCD	Vehicle (liposomes in water)
HCD + E	HCD	*H. sabdariffa.* + *Z. officinale* extract (1:1 for 8 days, after 2:1 for 45 days) in aqueous medium
HCD + L	HCD	Liposomal suspension (*H. sabdariffa. + Z. officinale* extract, 1:1 for 8 days, after 2:1 for 45 days)

STD: control; HCD: hypercaloric diet.

**Table 4 nutrients-17-02275-t004:** Polyphenol peaks identified in *Hibiscus sabdariffa* extract by FTICR-MS analysis. Mean intensity of three replicates.

ESI Mode	Measured *m*/*z*	mDa	Molecular Formula	Putative Annotation	Subclass
−	371.07382	−9.526	C15H14O11	2-*O*-Caffeoylglucarate	Phenolic acid
+	296.09862	−22.142	C13H13NO7	Caffeoylaspartic Acid	Phenolic acid
+	409.18447	−0.73	C21H18NO4	Chelerythrine	Phenolic acid
+	355.10342	−1.061	C16H18O9	Chlorogenic Acid	Phenolic acid
+	303.04859	−35.046	C14H6O8	Ellagic acid	Phenolic acid
−	321.04839	−24.282	C7H6O5	Gallic acid	Phenolic acid
−	135.00717	0.5	C7H6O4	Gentisic Acid	Phenolic acid
−	337.09506	−3.266	C16H18O8	P-Coumaroylquinic Acid	Phenolic acid
−	315.07377	−2.712	C13H16O9	Protocatechuic Acid 4-*O*-Glucoside	Phenolic acid
+	199.05286	7.24	C9H10O5	Syringic acid	Phenolic acid
+	338.07286	26.759	C16H17O8	Trans-5-*O*-(4-Coumaroyl)-d-Quinate	Phenolic acid
+	295.0512	39.8	C16H16O4	1,3-Cis-Tetrahydroxyphenylindan	Phenolic compound
+	433.40475	−0.743	C29H52O2	5-Tricosylresorcinol	Phenolic compound
+	133.06143	3.361	C9H8O	Cinnamaldehyde	Phenolic compound
+	349.12725	1.356	C16H24NO5	Sinapine	Phenolic compound
−	207.01611	12.69	C10H8O5	Fraxetin	Cumarin
−	321.04839	12.104	C15H14O8	(2*R*,3*S*,4*S*)-Leucodelphinidin	Flavonoid
+	317.0512	−22.006	C15H8O8	2,3,8-Trihydroxy-7-Methoxychromeno[5,4,3-Cde]Chromene-5,10-Dione	Flavonoid
+	433.14811	1.199	C22H24O9	3-Methoxynobiletin	Flavonoid
+	333.05927	1.224	C16H12O8	7-*O*-Methylmyricetin	Flavonoid
−	281.05333	−8.88	C16H12O6	Diosmetin	Flavonoid
+	451.96385	1.735	C17H13O8	Syringetin	Flavonoid
−	208.01927	4.82	C9H7NO5	Betalamate	Phenolic pigment

**Table 5 nutrients-17-02275-t005:** Polyphenol peaks identified in *Zingiber officnale* extract by FTICR-MS analysis. Mean intensity of three replicates.

ESI Mode	Measured *m*/*z*	mDa	Molecular Formula	Putative Annotation	Subclass
+	347.200029	9.081	C17H20NO3	(*S*)-Coclaurine	Benzylisoquinoline alkaloid
+	317.17328	−11.68	C18H22NO4	(*S*)-Norreticuline	Benzylisoquinoline alkaloid
+	343.09096	−9.731	C18H14O7	(*S*)-Usnate	Phenolic lactone
+	387.14257	−13.997	C17H22O10	1-*O*-Sinapoyl-Beta-d-Glucose	Phenolic glycoside
−	296.168	22.683	C17H28O4	6-Gingerdiol	Diarylheptanoid
−	299.23392	−3.024	C17H26O4	6-Gingerol	Alkylphenols (diarylpropane)
+	459.14236	−13.787	C23H22O10	6-*O*-Acetyldaidzin	Isoflavone glycoside
−	277.15738	22.441	C17H26O3	6-Paradol	Diarylheptanoid
+	375.15594	−7.65	C20H22O7	7-Hydroxymatairesinol	Lignan
+	373.16335	1.215	C21H24O6	Arctigenin	Lignan
+	111.0517	−7.644	C6H6O2	Catechol	Simple phenolic compound
+	369.13163	1.635	C21H20O6	Curcumin	Curcuminoid
−	269.08959	−8.755	C16H14O4	Echinatin	Chalcone
+	357.16865	−13.72	C21H24O5	Gingerenonea	Simple phenol
+	357.18337	7.01	C13H18O7	Guaiacol *O*-Beta-d-Glucopyranoside	Phenolic glycoside
+	361.17836	−13.795	C21H30O5	Humulone	Phenolic acid
+	389.15827	1.21	C21H24O7	Medioresinol	Lignan
+	403.13751	1.234	C21H22O8	Nobiletin	Flavonoid
+	387.16163	22.383	C15H18O8	P-Coumaric Acid 4-*O*-Glucoside	Glycosylated phenolic acid
+	247.05878	1.32	C13H10O5	Pimpinellin	Furanochromone
+	391.15222	−13.476	C20H22O8	Resveratrol 3-*O*-Glucoside	Glycosylated stilbene
+	471.14238	5.91	C28H22O7	Scirpusina	Lignan
+	275.14872	15.451	C17H24O3	Shogaol	Alkylphenols (diarylpropane)
+	371.14754	1.375	C21H22O6	Xanthohumol B	Flavonoid

**Table 6 nutrients-17-02275-t006:** Characterization of the liposomal suspension of *Hibiscus sabdariffa* and *Zingiber officinale*.

Total phenols (mg GAE/g dry sample)	18.44 ± 0.083
Particle size (nm)	153.46 ± 0.35
Polydispersity index	0.34 ± 0.02
Zeta potential (mV)	±0.62
Encapsulation efficiency of phenolic compounds (%)	64.4 ± 2.1
Concentration (particles/mL)	14,800 × 10^8^

**Table 7 nutrients-17-02275-t007:** Body weight variation at 8 and 45 days of treatment.

Group	8 Days of Treatment			45 Days of Treatment		
	Initial Weight	Final Weight	Gain or Loss Weight	Initial Weight	Final Weight	Gain or Loss Weight
STD	29.63 ± 2.57	31.52 ± 4.93	1.89 ± 2.35	36.34 ± 4.34	37.17 ± 4.78	0.82 ± 3.35
HCD + V	31.34 ± 2.86	34.75 ± 3.92	3.41 ± 1.05	38.17 ± 3.43	38.93 ± 4.43	0.75 ± 2.52
HCD + E	33.21 ± 3.1	25.54 ± 4.87	−7.66 ± 1.76	38.14 ± 3.49	33.15 ± 5.91	−4.98 ± 4.21
HCD + L	31.9 ± 2.75	26.99 ± 5.67	−4.91 ± 2.91	39.6 ± 3.41	38.43 ± 3.4	−1.15 ± 2.92

**Table 8 nutrients-17-02275-t008:** Biochemical analysis results and RFPW of animals from different treatments.

Variable	STD	HCD + V	HCD + E	HCD + L
Glucose (mg/dL)	85.2 ± 9.7 ^a^	104.5 ± 9.6 ^b^	115.6 ± 6.3	124.8 ± 5.1 ^b^
Tg (mg/dL)	63.17 ± 6.3	52.52 ± 7.4	69.66 ± 19	60.77 ± 8.2
TC (mg/dL)	101.86 ± 10.72 ^a^	158.69 ± 10.47 ^b^	132.13 ± 10.17	170.79 ± 8.4 ^b^
HDL-C (mg/dL)	20.23 ± 2.03 ^a^	17.88 ± 3.38 ^b^	25.61 ± 3.15	36.75 ± 3.93 ^c^
LDL/VLDL-C (mg/dL)	25.54 ± 4.95 ^a^	68.13 ± 7.35 ^b^	62.47 ± 7.49 ^b^	56.51 ± 7.67 ^b^
Atherogenic index	6.04 ± 2.71 ^a^	13.59 ± 4.16 ^b^	5.57 ± 1.54 ^a^	5 ± 1.37 ^a^
AST (mg/dL)	56.74 ± 2.54 ^a^	55.2 ± 4.38	51.6 ± 3.37	42.76 ± 3.82 ^b^
ALT (mg/dL)	12.32 ± 2.21	11.5 ± 2.49	11.81 ± 2.58	12.73 ± 1.05
RFPW(%)	1.34 ± 0.70 ^a^	3.03 ± 1.01 ^b^	2.33 ± 1.2	2.51 ± 0.62

Tg: Triglycerides; TC: Total Cholesterol; HDL-C: High-Density Lipoprotein Cholesterol; LDL/VLDL-C: Low-Density Lipoprotein/Very Low-Density Lipoprotein Cholesterol; AST: Aspartate Aminotransferase; ALT: Alanine Aminotransferase; RFPW: Relative Fat Pad Weigh; STD: Standard Diet; HCD: Hypercaloric Diet. The data represents the average ± the standard error. Different letters indicate significant differences between groups, Tukey post hoc, *p* ≤ 0.05.

## Data Availability

The original contributions presented in this study are included in the article. Further inquiries can be directed to the corresponding author.

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
