# Peer review of "Anti-Obesity Effect of Liposomal Suspension and Extracts of Hibiscus sabdariffa and Zingiber officinale in a Murine Model Fed a Hypercaloric Diet"

_nutrients, 2025, doi:10.3390/nu17142275_

Round 1

Reviewer 1 Report

Comments and Suggestions for Authors

The authors evaluated “the anti-obesity effect of liposomal suspension and extracts of 2 Hibiscus sabdariffa and Zingiber officinale in a mouse model fed a hypercaloric diet” as stated in the title.

This study is interesting, but I have several comments concerning the experimental procedure.

N=10 per group is a good number, but it is not specified whether the mice were housed individually. If not, there is a problem with food and water ingestion recording. Please describe precisely what was done.

Table 1: please indicate % mass and % kcal for fat, carbohydrate and protein.

Blood glucose and insulin values must be indicated.

Figures 3 and 6: the legend indicates the average food intake in calories. How do you explain the absence of differences between mice fed STD and HFD?

Figure 7: locomotor activity is normally reduced in HFD-fed mice compared with STD-fed mice. Any explanation?

Table 7: results of biochemical analyses of animals in the STD and HFD groups show no difference in ASAT, ALAT and cholesterol levels. Any explanation?

The English text must be read by a native speaker.

Comments on the Quality of English Language

The authors evaluated “the anti-obesity effect of liposomal suspension and extracts of 2 Hibiscus sabdariffa and Zingiber officinale in a mouse model fed a hypercaloric diet” as stated in the title.

This study is interesting, but I have several comments concerning the experimental procedure.

N=10 per group is a good number, but it is not specified whether the mice were housed individually. If not, there is a problem with food and water ingestion recording. Please describe precisely what was done.

Table 1: please indicate % mass and % kcal for fat, carbohydrate and protein.

Blood glucose and insulin values must be indicated.

Figures 3 and 6: the legend indicates the average food intake in calories. How do you explain the absence of differences between mice fed STD and HFD?

Figure 7: locomotor activity is normally reduced in HFD-fed mice compared with STD-fed mice. Any explanation?

Table 7: results of biochemical analyses of animals in the STD and HFD groups show no difference in ASAT, ALAT and cholesterol levels. Any explanation?

The English text must be read by a native speaker.

Author Response

Comment 1: N=10 per group is a good number, but it is not specified whether the mice were housed individually. If not, there is a problem with food and water ingestion recording. Please describe precisely what was done.

Response 1: We agree and specified in text line 165: Mice were individually housed in cages with sawdust bedding under temperature-controlled conditions (22 ± 1 °C), with 12-h light/dark cycles, with food and water ad libitum.

Comment 2: Table 1: please indicate % mass and % kcal for fat, carbohydrate and protein.

Response 2: we agree with the comment, we improve table 1 (line 187) by specifying the total percentage and what is equivalent in percentage to the total.

Macronutrients

Standard diet

STD

Hypercaloric diet

HCD

Proteins

0.909 cal/g (28.6 %)

0.841 cal/g (17.5%)

Carbohydrates

1.844 cal/g (58 %)

0.409 cal/g (40.9%)

Lipids

42.61 cal/g (13.4%)

2 cal/g (41.6%)

Total energy content

3.18 cal/g

4.81 cal/g

Comment 3: Figures 3 and 6: the legend indicates the average food intake in calories. How do you explain the absence of differences between mice fed STD and HFD?

Response 3: we agree, as mentioned in the paper, the mice show no difference in 8 days, due to the short time of the diet, however with 45 days of treatment (with longer time) they show significant difference (fig 5),

Comment 4: Figure 7: locomotor activity is normally reduced in HFD-fed mice compared with STD-fed mice. Any explanation?

Response 4: the explanation is: High-fat diet (HFD)-fed mice typically exhibit reduced locomotor activity compared to those fed a standard diet (STD). This decrease can be attributed to several interconnected factors. First, increased body weight and adiposity associated with HFD intake impose a greater mechanical load and physical burden, making movement more energetically costly and less efficient. Second, neurobiological alterations occur, including impaired dopaminergic signaling in reward and motor control pathways, as well as hypothalamic inflammation, which disrupts energy homeostasis and activity regulation. Additionally, HFD induces leptin resistance, blunting the hormone’s normal stimulatory effects on spontaneous physical activity. Lastly, systemic inflammation and metabolic fatigue may further contribute to reduced motivation and capacity for movement (Bjursell et al., 2008). We add a justification to the brief at line 344

Comment 5: Table 7: results of biochemical analyses of animals in the STD and HFD groups show no difference in ASAT, ALAT and cholesterol levels. Any explanation?

Response 5: the explanation is: The absence of differences in ALT and cholesterol levels between groups may be due to insufficient treatment duration to induce liver damage or lipid alterations. In addition, in early stages, liver damage may be mild or subclinical, and animals may activate compensatory mechanisms that prevent detectable biochemical changes. The specific composition of the diet and the genetics of the animal model also play a role, which may modulate the sensitivity to these effects. We have added the explanation with the above text in the discussion section line 411.

Reviewer 2 Report

Comments and Suggestions for Authors

This study aims to evaluate adjunctive therapies for obesity using extracts from Hibiscus sabdariffa (roselle) and Zingiber officinale (ginger). The extracts were prepared as regular suspensions and as liposomal suspensions, the latter being intended to enhance the activity of phenolic compounds and provide metabolic benefits. We believe that this research will contribute to improving our quality of life in the future. There appear to be no issues with the experimental methods or the results of the analysis.

On the other hand, please discuss the following points in more detail.

  1. This study is based on a mouse model, and its direct applicability to humans has not yet been verified. Considering differences in metabolism and immune response, it is unclear whether similar effects can be obtained in humans. Please expand the discussion on future prospects regarding this point.
  2. I feel that there is insufficient information regarding the evaluation of toxicity and side effects in long-term administration. Are there any concerns associated with the prolonged intake of these ingredients? Additionally, are there any measures to address and improve this issue?
  3. It seems that not only is there a lack of comparison between the simple extract and the liposome suspension, but also insufficient details regarding comparisons with commercially available anti-obesity drugs and the placebo group. Please deepen the discussion on applications for commercial products, utilization in pharmaceuticals, and to what extent this differs from previous approaches.

Author Response

Comment 1: This study is based on a mouse model, and its direct applicability to humans has not yet been verified. Considering differences in metabolism and immune response, it is unclear whether similar effects can be obtained in humans. Please expand the discussion on future prospects regarding this point.

Response 1:  we agree, added to the text (line 419): Hibiscus (Hibiscus sabdariffa) and ginger (Zingiber officinale) show a favorable safety profile in both animal and human studies. In rodents, hibiscus shows a high LDâ‚…â‚€ (>2,000-5,000 mg/kg), and in humans no significant adverse effects have been reported up to doses of approximately 10 g/day or extracts of 300 mg/kg/day (Hopkins et al., 2013). In addition, hibiscus has demonstrated beneficial effects on lowering blood pressure and blood lipid levels. Meanwhile, studies in rats indicate that oral doses of ginger up to 2,000 mg/kg/day are generally safe (Jeena et al., 2011), while in humans habitual consumption of 2-4 g/day is well tolerated, with mild, mainly gastrointestinal side effects (Bodagh et al., 2018). Ginger has shown anti-inflammatory properties, antioxidant properties, and positive effects in relieving nausea and digestive problems. These data suggest that the clinical uses of both products can be extrapolated to human use without significant risks of toxicity when therapeutic doses are respected.

Comment 2: I feel that there is insufficient information regarding the evaluation of toxicity and side effects in long-term administration. Are there any concerns associated with the prolonged intake of these ingredients? Additionally, are there any measures to address and improve this issue?

Response 2: we agree, added to the text (line 419): Hibiscus (Hibiscus sabdariffa) and ginger (Zingiber officinale) show a favorable safety profile in both animal and human studies. In rodents, hibiscus shows a high LDâ‚…â‚€ (>2,000-5,000 mg/kg), and in humans no significant adverse effects have been reported up to doses of approximately 10 g/day or extracts of 300 mg/kg/day (Hopkins et al., 2013). In addition, hibiscus has demonstrated beneficial effects on lowering blood pressure and blood lipid levels. Meanwhile, studies in rats indicate that oral doses of ginger up to 2,000 mg/kg/day are generally safe (Jeena et al., 2011), while in humans habitual consumption of 2-4 g/day is well tolerated, with mild, mainly gastrointestinal side effects (Bodagh et al., 2018). Ginger has shown anti-inflammatory properties, antioxidant properties, and positive effects in relieving nausea and digestive problems. These data suggest that the clinical uses of both products can be extrapolated to human use without significant risks of toxicity when therapeutic doses are respected.

Comment 3: It seems that not only is there a lack of comparison between the simple extract and the liposome suspension, but also insufficient details regarding comparisons with commercially available anti-obesity drugs and the placebo group. Please deepen the discussion on applications for commercial products, utilization in pharmaceuticals, and to what extent this differs from previous approaches.

Response 3: Answer: we agree, we add the necessary complement to the text (line 431): Drugs approved for the treatment of obesity act through specific mechanisms. These include inhibition of fat absorption (orlistat), activation of GLP-1 receptors to suppress appetite (liraglutide and semaglutide) and stimulation of the central nervous system to reduce hunger (phentermine). However, these drugs can cause relevant adverse effects, such as gastrointestinal disorders, cardiovascular and neuropsychiatric effects. This limits their short- or medium-term use under strict medical supervision. A recent analysis of 132 randomised clinical trials involving 48,209 participants found that the combination of phentermine and topiramate was the most effective in weight reduction, followed by GLP-1 agonists, with semaglutide showing the greatest benefit in terms of weight loss and risk of adverse effects (Shi et al., 2022).

In the present study, an effective reduction in body weight was observed with the administration of the simple extract combination (13.2 %) and with the liposomal suspen-sion (3 %) in a period of 45 days with a hypercaloric diet. In addition, in the context of the metabolic effect, the liposomal suspension treatment had better results compared to the other groups in terms of LDL/VLDL reduction and HDL increase, and this group of mice also remained more active and consumed more calories, suggesting that regardless of the damage caused by the DHC diet intake, the liposomal suspension had the capacity to attenuate its effects.As previously mentioned, coupled with the benefits of phenolic com-pounds present in hibiscus and ginger extracts, the use of oral liposomes in suspension form as an encapsulation medium is a specific alternative to enhance the effectiveness of various bioactives, which facilitate the arrival of bioactives to target organs and improve treatment performance (Jash et al., 2021).

Currently, there is no current evidence of the use of the combination of H. sabdariffa and Z. officinale extracts with anti-obesity effect, and even less with the use of liposomal formulations. There is only one study by Ahad et al. (2020) in which both extracts were administered with losartan to hypertensive rats. Treatment significantly reduced systolic and diastolic blood pressure. However, there is a great variability of studies individually of the extracts, with different models, doses, treatment times and results obtained, of which stand out the use of high doses and with therapeutic effects as in decreased food intake, decreased lipogenesis, increased lipolysis, stimulation of β-oxidation of fatty acids, inhibi-tion of adipocyte differentiation and growth, attenuation of inflammatory responses and suppression of oxidative stress (Maizatul et al., 2018; Morales-Luna et al., 2019; Sayed et al., 2020; Jason et al., 2021). The difference with the present study was that both extracts were mixed and formed liposomes for administration. The final effect was a decrease in body weight and some variables of the metabolic syndrome, in addition to reducing the liver enzyme AST, making it a potential hepatoprotective agent.

Round 2

Reviewer 1 Report

Comments and Suggestions for Authors

the MS is not at all clear what happened to the mice. It is imperative that the authors provide a flow chart explaining all the steps and that they use terms other than “8-day exposure” or “45 days on the high concentration” and “the remainder on the same assigned diet”.

They must provide data with body weight after the first 3 weeks of HFD/STD with the 40 mice and then how much extract/liposome was given for the first 8 days and how much was given the 45 following days 

Author Response

We agree with your comment and appreciate the observation. Please find the response in the attachment.
